# Remote Sensing System for Motor Nerve Impulse

**DOI:** 10.3390/s22082823

**Published:** 2022-04-07

**Authors:** Carmen Aura Moldovan, Marian Ion, David Catalin Dragomir, Silviu Dinulescu, Carmen Mihailescu, Eduard Franti, Monica Dascalu, Lidia Dobrescu, Dragos Dobrescu, Mirela-Iuliana Gheorghe, Lars-Cyril Blystad, Per Alfred Ohlckers, Luca Marchetti, Kristin Imenes, Birgitte Kasin Hønsvall, Jairo Ramirez-Sarabia, Ioan Lascar, Tiberiu Paul Neagu, Stefania Raita, Ruxandra Costea, Adrian Barbilian, Florentina Gherghiceanu, Cristian Stoica, Catalin Niculae, Gabriel Predoi, Vlad Carbunaru, Octavian Ionescu, Ana Maria Oproiu

**Affiliations:** 1IMT Bucharest, 77190 Bucharest, Romania; carmen.moldovan@imt.ro (C.A.M.); marian.ion@imt.ro (M.I.); david.dragomir@imt.ro (D.C.D.); silviu.dinulescu@imt.ro (S.D.); carmen.mihailescu@imt.ro (C.M.); eduard.franti@imt.ro (E.F.); octavian.ionescu@imt.ro (O.I.); 2ICIA, Ctr New Elect Architectures, 050711 Bucharest, Romania; monica.dascalu@upb.ro; 3Department DCAE, Faculty of Electronics, Telecommunication and Information Technology, University POLITEHNICA of Bucharest, 060042 Bucharest, Romania; lidia.dobrescu@upb.ro (L.D.); dragos.dobrescu@upb.ro (D.D.); mirelaiulianagheorghe@student.etti.upb.ro (M.-I.G.); 4Department of Microsystems, University of South-Eastern Norway, 3603 Horten, Norway; lars-cyril.blystad@usn.no (L.-C.B.); per.ohlckers@usn.no (P.A.O.); luca.marchetti@usn.no (L.M.); kristin.imenes@usn.no (K.I.); birgitte.honsvall@usn.no (B.K.H.); jairo.ramirez@usn.no (J.R.-S.); 5Emergency Clin Hosp Bucharest, 014461 Bucharest, Romania; ioan.lascar@umfcd.ro (I.L.); tiberiu.paul@mfcd.ro (T.P.N.); vlad.carbunaru@umfcd.ro (V.C.); 6Department I, Faculty of Veterinary Medicine, University of Agricultural Sciences and Veterinary Medicine of Bucharest, 050097 Bucharest, Romania; stefania.raita@fmvb.ro (S.R.); ruxandra.costea@fmvb.ro (R.C.); catalin.predoi@fmvb.ro (G.P.); 7Orthopedics, Anaesthesia Intensive Care Unit, Faculty of Medicine, UMF Carol Davila, 050454 Bucharest, Romania; adrian.barbilian@umfcd.ro (A.B.); florentina.gherghiceanu@umfcd.ro (F.G.); cristian.stoica@umfcd.ro (C.S.); 8AREUS Technol SRL, 70000 Bucharest, Romania; catalin.niculae@areusdev.com; 9Department IME/Faculty ACE, Petroleum and Gas University from Ploiesti, 100680 Ploiesti, Romania

**Keywords:** neuroprosthesis, microtechnology, microsystem, microfabrication

## Abstract

In this article, we present our research achievements regarding the development of a remote sensing system for motor pulse acquisition, as a first step towards a complete neuroprosthetic arm. We present the fabrication process of an implantable electrode for nerve impulse acquisition, together with an innovative wirelessly controlled system. In our study, these were combined into an implantable device for attachment to peripheral nerves. Mechanical and biocompatibility tests were performed, as well as in vivo testing on pigs using the developed system. This testing and the experimental results are presented in a comprehensive manner, demonstrating that the system is capable of accomplishing the requirements of its designed application. Most significantly, neural electrical signals were acquired and transmitted out of the body during animal experiments, which were conducted according to ethical regulations in the field.

## 1. Introduction

Important developments in the prosthetics industry are continuously being reported, revealing, in particular, a new research trend towards the creation of a robotic arm capable of using nervous electrical signals for its control and feedback, capturing motor impulses, and transmitting feedback to peripheral nerves [1]. From a practical perspective, an arm prosthesis is expected to act similarly to the limb it replaces, but this depends on the type of signals acquired from the patient’s stump and used for the prosthesis movement. Myoelectric prostheses use signals acquired from the stump muscles but can only make a small number of movements. However, neural prostheses use signals acquired from the stump’s nerves, which allows patients to perform more movements with higher accuracy. Thus, the architecture of the prosthesis must include a high-performance system capable of acquiring the signals from the patient’s stump. The electrodes presented in this article were specially designed to acquire, with a high resolution, neural electrical signals from the median and ulnar nerves in the patient’s stump. They are flexible collar electrodes, which are included in the category of cuff electrodes because they are mounted circularly around the nerves; however, ours have a different design with an additional reference electrode that allows for the acquisition of both alternating motor neural signals [2].

Cuff electrodes were patented and fabricated for the first time in 1984 by Naples et al. [3]. Since then, they have been used in implantable interfaces for many types of medical applications [4]:The stimulation and recording of neural activity in the peripheral nervous system [5];The acquisition of neural electrical signals to control motor prostheses [6,7];The stimulation of the optic nerve for visual prostheses [8];The electrical stimulation of nerves for the restoration of motor functions [9].

The electrode we designed consists of two thin, flexible pairs of gold electrodes on a 25 micron thick polyimide substrate; in this paper, it can be wrapped around the nerves, so we call it a flexible collar electrode. Since the nerve impulse has both negative and positive components, an additional reference electrode was designed. The reference electrode is intended to be implanted in the body at a neutral point, preferably near a bone joint. 

Nerve pulse propagation is described in [10] by A. L. Hodgkin and A. F. Huxley. The shape of the pulse is generally accepted as that presented in Figure 1. The electrical signal acquired from the nerve usually has an amplitude in the millivolt range; therefore, it requires pre-amplification and filtering before further processing.

The remote sensing system for motor nerve impulse acquisition introduced in this article is shown in Figure 2. The front-end module provides both filtering and an amplification of the nerve pulse to the level required by a microcontroller. The signal is further digitized and wirelessly transmitted via a Bluetooth Low Energy (BLE) module to a mobile phone. The implantable electronic module is powered by a lithium-ion battery.

Because they are used as implantable devices, cuff electrodes must be designed exclusively with soft, biocompatible materials to avoid any damage of the surrounding nerve/tissue, inflammation, infection, or other rejection reactions of the body. They must have the appropriate mechanical, physical, and biochemical properties to attenuate the human body’s reaction to foreign bodies and they should maintain these properties for a long time after implantation [11,12,13] in the human body’s conditions (high level of oxygen, macromolecules, and saline electrolytes in the tissue). Destructive effects include the detaching of the electrodes from the substrate, the corrosion of the metal lines, and/or the breaking of the flexible substrate [14]. 

Various authors recommend polymers as the substrate material, and gold, platinum, tungsten, or stainless steel for the electrodes [15,16,17,18,19]. Other options for the electrodes, reported in recent articles, are iridium, tantalum pentoxide, and titanium nitride [20,21]. There are no general rules for the fabrication of cuff electrodes, as they must be adapted specifically to the type of tissue in which they are to be implanted. We carefully considered fibroblasts, which usually appear after surgery, because of the possibility of them covering all parts of the electrodes and thus affecting their proper functioning [22]. After implantation of the cuff electrodes, damage to the nerves or tissues and the risk of neuroinflammatory reactions were monitored [22].

## 2. Materials and Methods

### 2.1. Flexible Collar Electrodes

The concept for our electrodes is based on the design and fabrication of a system of flexible electrodes that can be attached to nerve bundles to receive or transmit the existing difference in electrical potential when transmitting or receiving a stimulus. The electrodes were composed of a flexible polyimide substrate with a conductive chromium-gold (Cr-Au) layer with a thickness of 20 nm Cr and 200 nm Au. The role of the chromium layer was to ensure the adhesion of the gold onto the polyimide substrate. The Cr/Au conductive layer was deposited using the sputtering technique on a 0.025 mm thick KAPTON insulating foil, thus ensuring that the basic requirements imposed for the developed application were met, i.e., biocompatibility, high electrical conductivity, and flexibility. The electrodes were designed in a way to allow them to wrap around the nerve bundle. They were given holes through which sutures could be made to ensure firm electrical contact with the bundle.

The novelty of our proposed sensor is found in its unique technological flow, which has strong advantages, as it uses cheap, flexible, and biocompatible materials, and has significantly lower costs than existing methods. 

Figure 3 shows the method and technological steps in the electrode fabrication.

The Kapton foil was fixed onto a glass support, followed by the photolithographic process (M1). A photoresist layer (1.2 microns) was deposited by spinning onto the Kapton foil, soft baked (90 °C), exposed to UV through the mask for electrode fabrication (M1), developed, rinsed, and dried.

After drying, the Kapton was shortly exposed to an oxygen plasma to better clean and dry the surface in order to ensure the adhesion of the metal layer. The assembly was then deposited into 20 nm Cr/200 nm Au, using the Edwards sputtering system. Then, the photoresist was removed in acetone heated to 50 °C and left for 30 min in an ultrasonic bath, leading to the removal of any remaining photoresist (lift-off). The conductive paths designed to be in contact with tissue were coated with an insulating layer of PDMS by spin coating and using a mask (M2), leaving open windows for pads and contacts to the nerve. After this, the electrode system was exposed to temperatures of around 80 °C for a treatment of 120 min in a nitrogen atmosphere. After fabrication, the reference electrodes were created using precise laser cutting. Finally, the sutures holes were made and the edges removed using computer-controlled CO_2_ laser cutting equipment. The design of the electrodes is shown in Figure 4. 

### 2.2. Electronic Circuit

A testing circuit was developed to verify the effectiveness of the fabricated electrodes. The first module was the front-end circuit necessary to amplify the motor nerve pulse signal, which has an amplitude of approximately 5–10 mV. 

The first step of the amplification process was to set a gain of G1 = 100 to minimize subsequent noise contribution. Considering that the nerve pulse has both a positive and a negative component, the circuit used for this first step was the INA118, which is a low-power, general-purpose instrumentation amplifier offering excellent accuracy. Its versatility, design, and small size (3.91 mm × 4.90 mm for SOIC version) make this device an excellent choice for a wide range of applications. Its current-feedback input circuitry provides wide bandwidth, even at high gain (70 kHz at G = 100). The INA118 operates with power supplies as low as ±1.35 V and the quiescent current is only 350 µA, making this device well suited for battery-operated systems. While our in vivo objective was a 5 mVpp amplitude stimulus, the potential of the body to react to foreign bodies by creating scar tissue and thus raising impedance over time [22,23,24,25] means that the amplitude of the input signal could decrease accordingly. In order to develop a versatile front-end for the used electrodes, we decided to choose a gain of 100 for the INA118 and to use a chain of two amplifiers to establish a control loop on the second amplifier, in order to compensate for this variation. We look forward to performing a long-term test as soon as a solution for the wireless charging of batteries has been developed, and then, based on the obtained data, we will increase the gain of INA118 if necessary or compensate for the impedance modification only by maintaining the already established control loop. We looked also for an initial solution providing flexibility and that was easy to modify without changing the PCB design, or in the worst case, a complete modification of electronic design.

The second stage of amplification was completed on an operational amplifier OPA2340 of small dimensions (OPA2340 chassis type VSSOP 3.00 mm × 3.00 mm). The OPA340 series operates on a single supply as low as 2.5 V with a common-mode input voltage range that extends 500 mV below the ground and 500 mV above the positive supply. Its output voltage swing is within 1 mV of the supply rails for a 100-kΩ load. These devices also offer an excellent dynamic response. The first half of the circuit was used as a voltage follower and the second half as an amplifier with a gain of G2 = 10. The value of 0.5 V was not considered an acceptable value for the pulse amplitude and it was thus raised to approximately 5 V, a level usable for the input of any controller.

Diagrams of the designed front-end circuit and of the circuit itself are shown in Figure 5 and Figure 6.

Intended to be used for neuroprosthesis, our system only needs a small range of wireless signal propagation, as the control block of the prosthesis is located in proximity to the stump. However, in the animal experiment presented in Section 3.4, this system was implanted in a swine leg, without amputation. The acquired neural electrical signals were processed in vivo and wirelessly transmitted out of the animal’s body. Therefore, for the experimental phase, there was a need to increase the length of the wireless link initially designed to be established between the receiver module located in the prosthetic device and the emitter embedded in the patient’s arm. Consequently, for the experiments on swine, we decided to use devices based on the Bluetooth protocol, which would provide a suitable operational environment, as they could be connected easily to mobile devices. In addition, using these modules provided us the opportunity to develop a more reliable demonstrator and focus on the electrodes and front-end module, which were newly designed and developed. Thus, an ARDUINO Mini module, an HC-05 Bluetooth shield, an iOptron Lithium-Poly 3.7 V battery, and a 2 V to 5 V step-up DC-DC converter, as shown in Figure 7, were our choice for the development of the communication system. The modules were encapsulated in PDMS to ensure the required biocompatibility.

### 2.3. Cytotoxicity Tests of Electrodes

Cytotoxicity tests of three electrodes were performed by investigating cell growth, measuring metabolic activity, and counting cells after the electrodes had been maintained for 28 h in a cell culture. The motivation for the cytotoxicity testing was to determine the cytocompatibility of the fabricated electrodes and whether they were suitable for in vivo use. The components tested were the Cr/Au/Kapton electrodes.

The electrodes were sterilized by ultrasonication in 70% ethanol and the cytotoxicity tests were carried out using a mouse osteoblastic cell line (MC3T3-E1, European Collection of Authenticated Cell Cultures). The cell culture was cultivated at 37 °C in a humidified atmosphere of 5% CO_2_, in Minimal Essential Medium Eagle (Sigma-Aldrich, St. Louis, MO, USA) with 10% fetal bovine serum (Gibco) and antibiotics (100 IU penicillin/mL and 100 µg streptomycin/mL, Gibco). After two days of growth, the cells were transferred into T25 culture flasks, where the cell culture was incubated for 24 h before introducing the components to be tested. We tested positive controls, negative controls, and the electrodes themselves; all three variants were tested in triplicates. No components were added to the positive controls; a piece of copper wire was added to the negative controls; and flexible collar electrodes were added into the flasks for the “electrode” tests. A fresh growth medium was added and the cell culture was incubated for another 24 h. Cell growth was then investigated under a microscope; metabolism was measured fluorescently after an addition of PrestoBlue (Invitrogen, Waltham, MA, USA) and cells were counted in a hemocytometer. Statistical analysis was performed using ANOVA and the unequal variance *t* test (α = 0.1).

### 2.4. Corrosion Tests 

Implantable electrodes used in the medical field can be characterized in terms of their electrochemical, mechanical, and surface properties. Stimulation electrodes, especially, can cause electrochemical problems due to the high current density they discharge. These electrodes are surrounded by tissue and electrolytes; thus, they are eventually stimulated, resulting in electric fields and induced polarizations that change the local ion concentrations and local pH and can cause problems such as corrosion of the material and fouling of the electrode.

Changes in resistance can be caused by adsorption of ions from electrolyte solutions onto the electrode, by electrode corrosion, by an encapsulation of the electrode in the fibrous tissue (known as glial scarring), or changes in the chemical environment around the electrode. Unwanted polarization of the electrode can lead to adverse effects such as corrosion, dirt, and toxicity. Equilibrium potential, pH, and required current density should all be considered when choosing the material for the electrode, as these may affect the chemistry of the interface.

Electrochemical techniques, electrochemical impedance spectroscopy (EIS), and cyclic voltammetry (CV) were used to characterize the gold electrode on the Kapton. EIS is a method that evaluates the resistance of the electrode through the AC method and is represented by Nyquist diagrams when the impedance is a plot of Z_imaginary_ vs. Z_real_, as a function of frequency. A 1 kHz impedance is usually the typical frequency for neural stimulation, and the bulk resistance of the implantable electrode is calculated using an extrapolation of the Nyquist curve by finding the *X*-axis intercept in the high-frequency region of the Nyquist diagram [26]. Nyquist diagrams (-Zi vs. Zr) were recorded for a frequency range between 100 mHz and 100 kHz, specifically following the 1 kHz impedance, the bulk impedance of the implantable electrode, and the diameter of the semi-circle that usually represents the charge transfer impedance. It is important to evaluate the electrode using EIS because it can provide data on the impedance of the electrode and the electrode interface’s behavior in relation to the biological medium.

The electrolyte solution used was pH 7.1 PBS (phosphate buffered saline), the standard solution for physiological medium experiments. The electrochemical experiments were performed using VoltaLabPGZ100. The gold electrodes were used as the working electrode, while the counter electrode was Pt and the reference electrode was Ag/AgCl (3 m KCl). The electrochemical impedance spectroscopy (EIS) experiments were performed at an open circuit potential in the frequency range of 100 kHz and 0.1 Hz with a 10 mV amplitude, and PBS 7.1 was the supporting electrolyte solution. The EIS recordings were recorded for 21 days. Cyclic voltammetry (CV) is a characterization technique used to investigate the stability of the electrode surface by verifying if there are oxido-reduction peaks and to check for possible adsorption processes that may occur after application of an alternating voltage. The CV experiments were also performed using VoltaLabPGZ100 with a potential swiped between −800 mv to +800 mV after immersion of the electrode during the 21st day in a PBS 7.1 electrolyte solution with a rate of scan of 50 mV/s.

### 2.5. Mechanical Tests

Implanted electrodes experience a high strain because muscle movements lead to their repeated bending. Therefore, we conducted mechanical tests consisting of repeated cycles of bending the electrodes using the MECMESIN Multitest 2.5i shown in Figure 8a. The cycling program was set with the following parameters: minimum displacement = 0, maximum displacement = 50 mm, speed of displacement = 10 mm/min, landing times = 0.1 min. A total of 3 tests, of 50 bending cycles each, were performed on the test electrodes, as shown in Figure 8b. After each test, the electrodes were observed with an optical microscope to examine whether a segregation of the gold layer or any micro-cracks/interruptions of the circuits occurred. 

The resistance of the electrodes was measured for a set of 8 electrodes before and after bending. A fragment of the tested set is shown in Figure 9. 

The measurements were conducted using a FLUKE 8846 A multi-meter, in configuration 4 wires, as shown in Figure 10. 

### 2.6. In Vivo Tests on Developed Electrodes

In vivo experiments were necessary to validate the design of the electrodes, as well as the behavior of the implantable modules in real-life conditions. Due to the complexity of the electronics and the high number of unknown situations which could occur, it was clear to us that a procedure involving verifications should be conducted at each step of surgery, as well as during the assembly of the electrodes with the front-end and wireless systems.

In order to fulfill these requirements, the experiment was planned to be conducted over a period of three days on two subjects, thus having a contingency plan for unexpected situations. The study included two specimens of Sus scrofa domesticus (domestic pig), weighing around 50 kg each. We used the swine experimental model for its remarkable resemblance to that of humans in terms of nerve diameter, conduction, and impulse transmission. To reduce stress, the swine were accommodated in the experimental research facility and monitored for 48 h before the interventions. Free access to water was allowed, while feeding was stopped 18 h before anesthesia. All the procedures were approved by the Faculty of Veterinary Medicine Ethics Committee and adhered to the guidelines outlined in the Public Health Service Policy on Humane Care and Use of Laboratory Animals (2015). All the surgical procedures were performed with the swine under general anesthesia. We commenced with the rapid administration, in the cervical area of the trapezius muscle, of intramuscular premedication using Ketamine 20 mg/kg and Xylazine 1 mg/kg for deep sedation and immobilization. After 10 min, venous access was performed using an 18-gauge catheter, in the left lateral auricular vein. As for the induction of anesthesia, it was administered intravenously using Propofol 3 mg/kg, followed by intubation using a long blade laryngoscope, a stylet, and a 5.5 tube. After obtaining an appropriate level of anesthesia, the experimental animal was positioned so as to facilitate the approach to the deep peroneal nerve.

According to the scenario proposed on the first day of experiments, the standard procedure for swine surgery was conducted (anesthesia, operation field preparation, etc.). The surgeons decided on the best place for the incision and then the procedure for unveiling the nerve was conducted. After the nerve was revealed, the flexible collar electrode was installed on the nerve as shown in Figure 11, and the testing operations were performed.

All the procedures from the animal experiments were approved by the Faculty of Veterinary Medicine Ethics Committee and by Romanian Sanitary Veterinary Directorate and adhered to the guidelines outlined in the Directive 2003/65/EC of the European Parliament and of the Council of 22 July 2003 amending Council Directive 86/609/EEC on the approximation of laws, regulations, and administrative provisions of the Member States regarding the protection of animals used for experimental and other scientific purposes [27]. 

## 3. Results and Discussion

### 3.1. Results of Cytotoxicity Tests of Electrodes

Inspection of cell growth indicated no visual difference between the positive controls and cultures with the flexible collar electrodes. Metabolic activity did not indicate any difference between the positive controls and electrodes, as shown in Figure 12. Inspection of the negative controls showed cell death near the copper wire, but further away from the copper the cells seemed to grow as normal. Mean cell counts were 1.61 × 10^6^ cells/mL (SD 4.8 × 10^5^ cells/mL) for the positive controls; 6.2 × 10^5^ cells/mL (SD 1.06 × 10^5^ cells/mL) for the negative controls; and 1.24 × 10^6^ cells/mL (SD 1.71 × 10^5^ cells/mL) for the flexible collar electrodes. There was no statistical difference in cell numbers between the positive controls and the flexible collar electrodes; however, there was statistical difference in cell numbers between the negative controls and the other groups (α = 0.1). 

The local effect of the copper wire may explain why the difference between the negative and positive controls was relatively small both for metabolic activity and cell numbers. However, there still were significant differences between the negative copper controls and the electrodes, both in cell numbers and metabolic activity. This indicates a low cytotoxicity of the flexible collar electrodes.

### 3.2. Results of Corrosion Tests

The inset in Figure 13 shows the equivalent circuit model for the electrode/PBS electrolyte interface. The circuit consists of a capacitive interface of the double layer in parallel with the charge transfer resistance (Rct) and in series with the solution resistance (Rs). The stability of the electrodes was determined by immersing them in a PBS solution over the course of 21days and, throughout this period, recording the Nyquist plots (-Zi vs. Zr) from impedance spectroscopy at 0, 1, 7, 14, and 21 days.

Figure 14 shows that, from the beginning of the measurement, the electrode presents a high conductivity; the semicircle of the Nyquist plots is not observed because there is no existing Rct at the interface until the 7th day. After this period, the resistance to electronic transfer can be observed in the 14th and 21st day. Some ionic processes that take place at the surface of the electrodes can be observed by recording cyclic voltammograms (CVs). Figure 14 shows an oxido-reduction transfer at the electrode interface that increases with time, probably due to the gold electrode’s interaction with the electrolyte ions, or an oxidation of chloride ions or phosphate ions.

This process may create instability over time, which could lead to electrode degradation, with the formation of an oxide layer on the electrode that could affect the charge delivery capacity [28,29]. 

To avoid these disadvantages, it is recommended to cover the gold with a matrix of stable biocompatible polymers that will not affect the chemistry of the interface over time. Also, from the impedance spectra shown in Figure 14, the increase of the Rct values after two weeks shows that there is a coating of the electrode surface over time, probably due to the deposition of some salts present in the electrolyte solution. However, the gold, compared to other electrodes (Pt, Ti), offers one of the smoothest surfaces and also a more hydrophobic surface [30]. 

Figure 15 presents the logarithm of impedance spectra for gold on the 21st day of experiments (a–e) versus the logarithm of frequency. All impedances increase with the decrease of frequency. The frequency range of neuromuscular electrical stimulation (NMES FR) is described, in other references, to be between 10 Hz and 1000 kHz [24]. At low frequencies, the capacitive double layer became negligible and the circuit thus became Rct + Rs. The solution resistance was very low compared to Rct, and the impedance changes at low frequencies (below 1000 Hz) were mainly influenced by the ambient conditions and by the roughness of the electrode. 

The range of NMES FR between 10 and 1000 Hz contains a cut-off intercept of rising slope tangent with constant impedances. This is called the cut off frequency (fc) and is described as the charge transfer ability of the gold electrodes to the electrolyte (or to the biological environment for in vivo experiments). We observed a cut-off frequency at about 1000 Hz that decreased with the increase of the immersion time, but the gold electrode showed a consistency of slope over time. The decrease of slope, or Rct, is a measure of low capacitive behaviour of the electrode over time [30]. 

An electrode with a rougher surface may also determine a lower cut off frequency due to its increased area. The roughness of the gold surface, over time, is modified most likely due the adsorption of water ions, but the cut-off frequency in the case of our gold electrode’s immersion in PBS solution over time is stable compared to other electrode materials that could have a hydrophilic surface, as has been shown in other studies [30,31,32].

The impedance at 1 kHz decreased from 83 ohms to 53 ohms, as is shown in Figure 16, probably due the PBS solution that contains NaCl within the electrode and their ionic conductivity (observed also by CVs behaviours over time), which led to a slight decrease of the electrode impedance. However, this is good for the proposed application because the impedance is low for the measuring of a clear signal with very low noise.

### 3.3. Results of Mechanical Tests

The results are presented in Table 1 and in Figure 17 and indicate that there was no physical damage to the electrodes.

### 3.4. Results from In Vivo Tests on Developed Electrodes

The goal of the first test was to check the connections, as short circuits between the terminals could have been caused during the medical procedure for the attachment of the flexible collar electrodes. For this test, the flexible collar electrode was connected to a specially manufactured testing device (cable with receptacle connector) and the resistance between the two pairs of terminals was verified. The measurements proved that there were no short circuits. The second step of verification was to ensure that the flexible collar electrode was in contact with the nerve. This step was performed as shown in the block diagram in Figure 18. 

A pulse generator was used to generate rectangular pulses (0.1 V, 1 kHz) to stimulate the nerve. According to available information in the scientific literature, the maximum frequency of neural impulses is estimated to be around 1 kHz [22,23,24,25,33] and therefore, the tests were conducted at 1 kHz. When applying the signal to the nerve, as is shown in Figure 19, it was observed that the leg of the swine was moving.

After this step, we ensured that there was good contact between the electrode and the nerve, the adapter was disconnected, and the front-end module was connected to the flexible collar electrode, as shown in Figure 20.

After the connection of the front-end module, a test was conducted to verify the interface connection with the flexible collar electrode, and to check that the preamplifier was able to receive a proper electrical signal from the flexible collar electrode. A pulse generator stimulated the nerve and the received signal at the preamplifier output was recorded using an oscilloscope. The test also confirmed that the preamplifier output had the right value to trigger the microcontroller input. The block diagram of the experiment is shown in Figure 21.

The last step of the verification was to establish and validate the entire chain of electronics, and to record nerve pulses through wireless communication (Bluetooth). The block diagram of the electronic module is shown in Figure 22.

To avoid any interferences, the stimulus was made at a level of 0.01 V by touching the nerve to the terminal of the pulse generator set on the step. The recorded signal on the mobile phone (Figure 23) demonstrated that the system was working, and the impulse was transmitted to the receiver.

A series of signals with 1 kHz frequency and 0.01 V amplitude was applied to the nerve. The electrical signals acquired from the nerve through the electronic test module and wirelessly transmitted to the mobile device are presented in Figure 24.

## 4. Conclusions

The goal of this research is to provide a reliable and stable neural interface for limb prosthesis, using wireless communication (BLE) between implanted electronic modules and external mechatronic structures. In this study, we present a remote sensing system for motor nerve impulse acquisition, consisting of flexible collar electrodes, an implantable electronic front-end circuit including wireless transmitter, and an external receiver block. Several cytotoxicity, performance, and reliability tests showed that the electrodes are reliable and effective for this application. The functionality of the whole system was demonstrated by successfully recorded sent and received nerve motor electrical signals during ethically conducted animal experiments. 

The experimental results show that the electrodes’ architecture is adequate for this application, as it improves the reliability of electrodes by using a redundant pair and the sensitivity using a differential amplification module and a reference electrode. The repeated bending tests demonstrated the electrodes’ excellent mechanical properties. However, the appearance of redox processes over time and the instability manifested by an increase in the electrical resistance, correlated with a decrease in conductivity over 21 days of immersion in electrolyte solution, showed an electrochemical instability which could lead to degradation over time. 

The results obtained so far and presented in this paper prove that the design of the developed sensing system is adequate to the targeted application. During the animal experiments presented, we were able both to detect motor impulses induced artificially and to stimulate the leg muscles through the flexible collar electrodes; however, we noted that complex applications, such as a patient stump implantation, where motor nerve impulses must be acquired from at least four different nerves, will require additional developments in the electrode system and processing circuitry. Further steps in this research also include finding solutions to increase the electrochemical stability of the electrodes and long-term in vivo experiments.

## Figures and Tables

**Figure 1 sensors-22-02823-f001:**
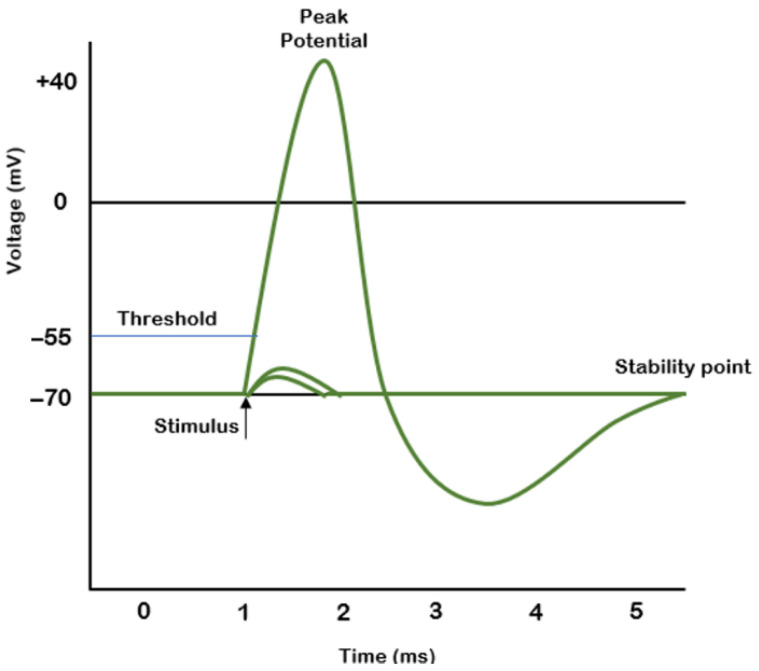
The shape and characteristics of a motor nerve impulse.

**Figure 2 sensors-22-02823-f002:**
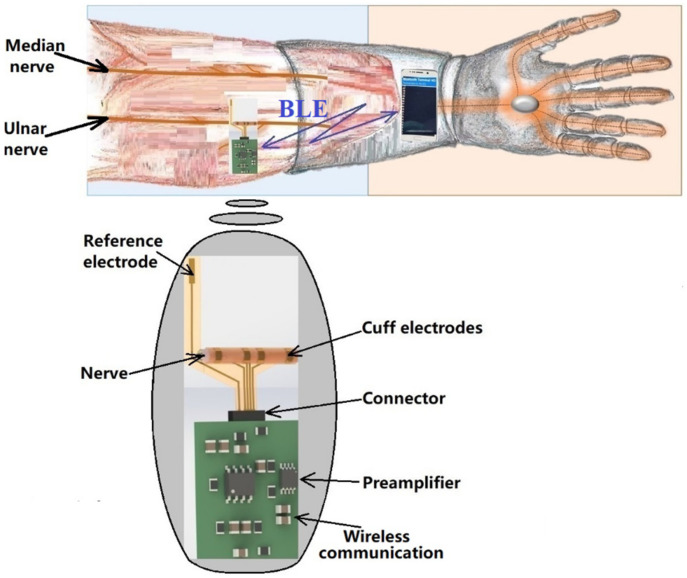
The remote sensing system for motor nerve pulse acquisition.

**Figure 3 sensors-22-02823-f003:**
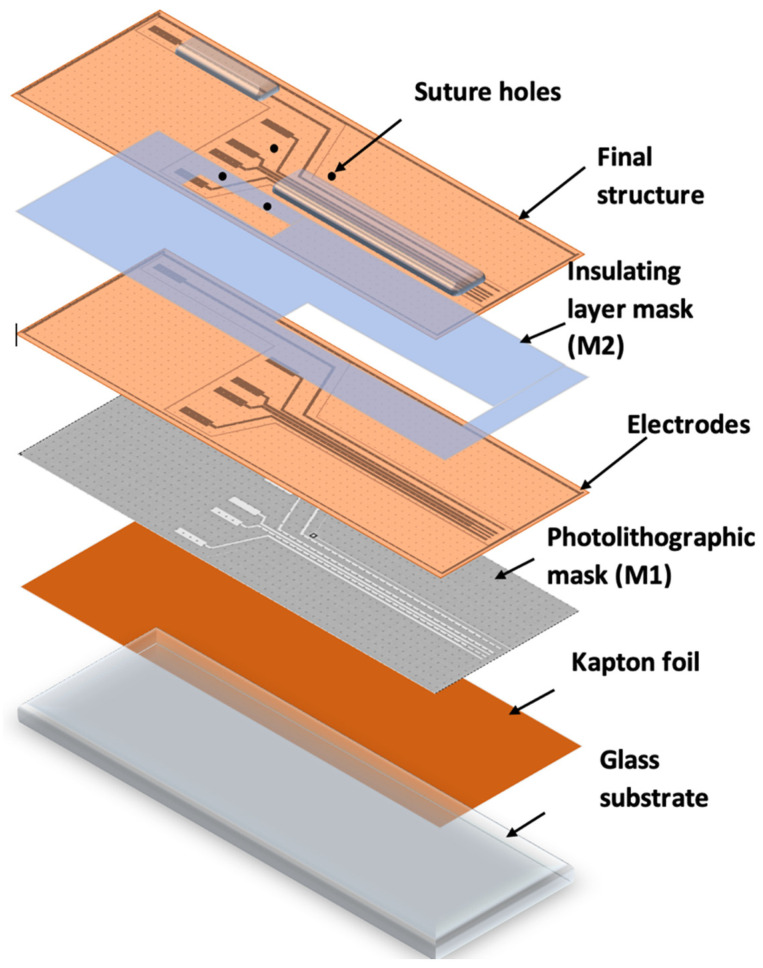
Schematic view of the method and the technological flow of the electrode system.

**Figure 4 sensors-22-02823-f004:**
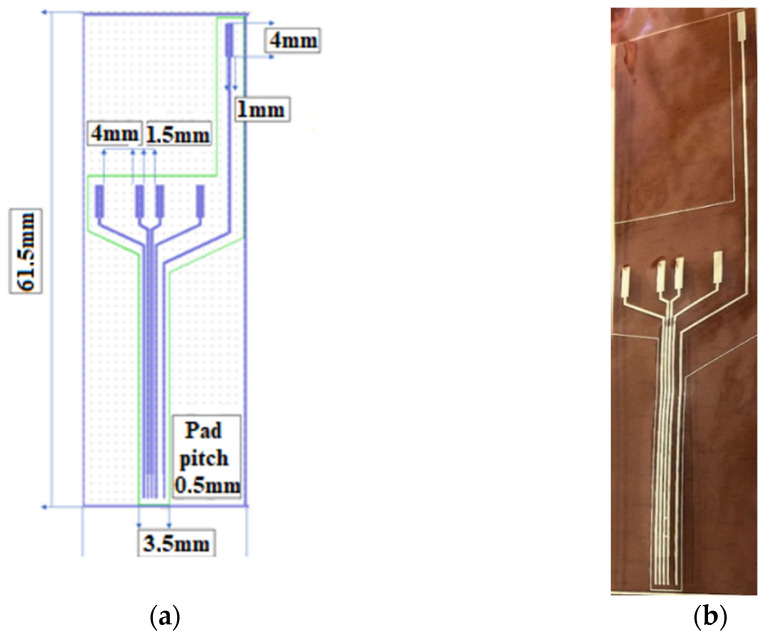
(**a**) Layout of the proposed flexible electrodes and (**b**) fabricated electrodes.

**Figure 5 sensors-22-02823-f005:**
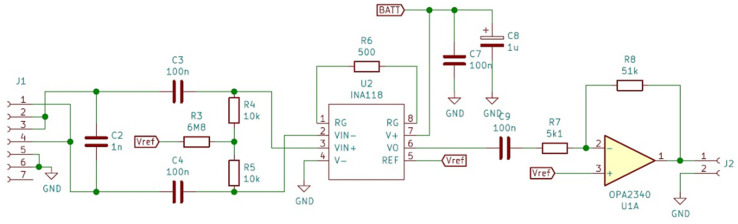
The circuit diagram of the front-end module.

**Figure 6 sensors-22-02823-f006:**
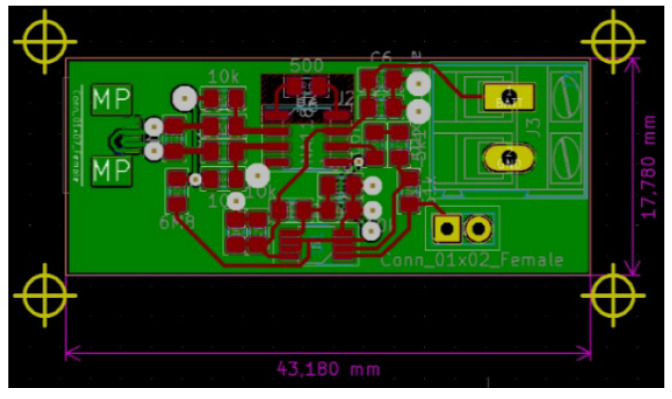
The front-end circuit board.

**Figure 7 sensors-22-02823-f007:**
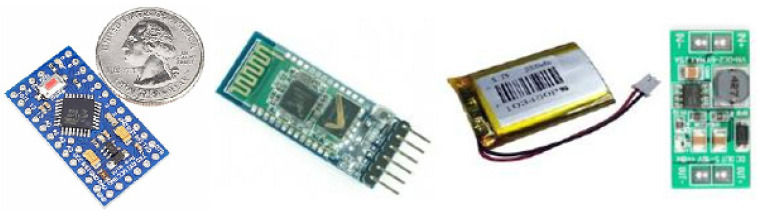
Arduino Mini, HC-05 Bluetooth module, iOptron Lithium-Poly 3.7 V battery, and the step-up DC-DC converter.

**Figure 8 sensors-22-02823-f008:**
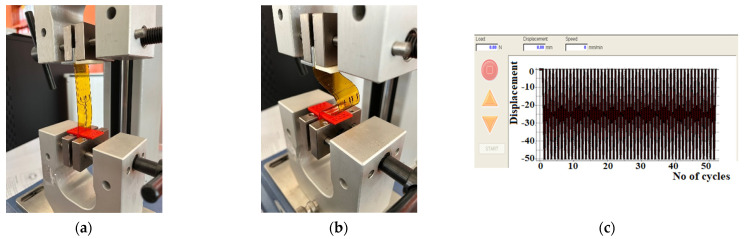
The bending cycles: (**a**) stretching and (**b**) bending the electrodes with MECMESIN MULTITEST 2.5.i; (**c**) diagram of the bending cycles.

**Figure 9 sensors-22-02823-f009:**
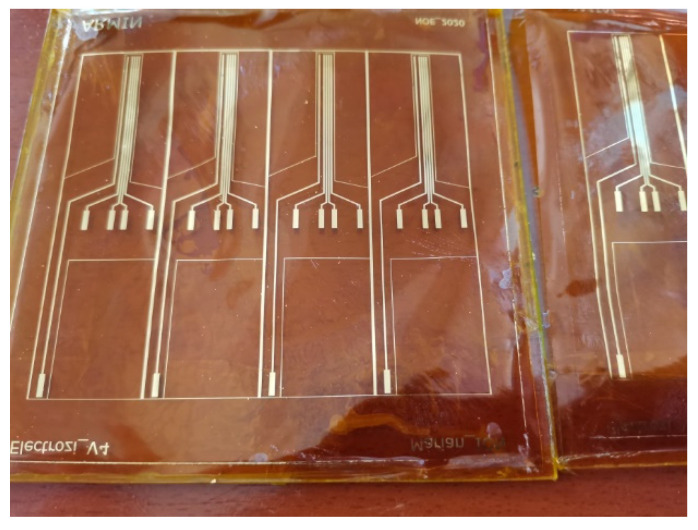
Electrodes used as samples for the bending tests.

**Figure 10 sensors-22-02823-f010:**
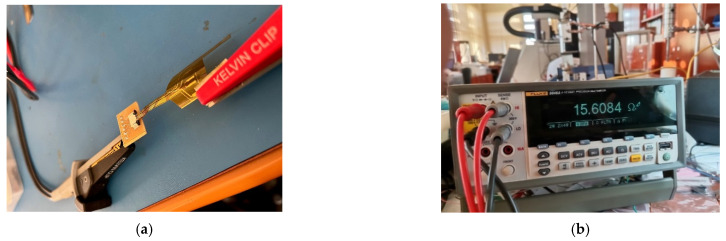
Measurements the electrodes resistance: (**a**) electrodes (**b**) FLUKE 8846 A multi-meter.

**Figure 11 sensors-22-02823-f011:**
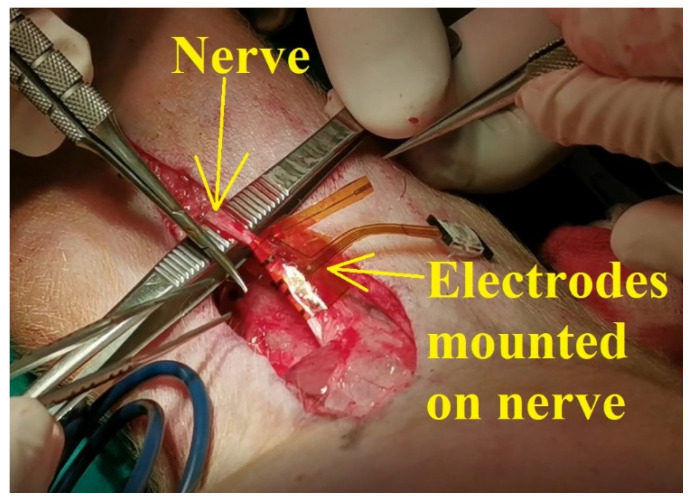
The flexible collar electrodes attached to the nerve.

**Figure 12 sensors-22-02823-f012:**
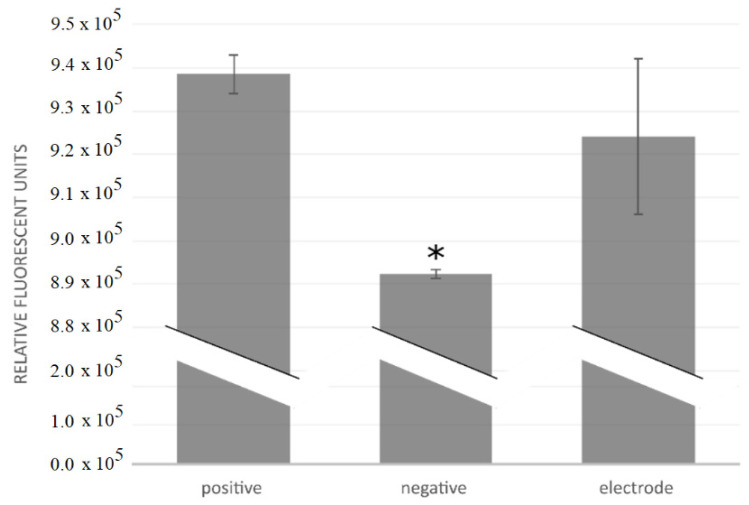
Metabolic activity measured fluorescently after addition of PrestoBlue reagent. An * denotes statistical difference from the other groups (α = 0.1).

**Figure 13 sensors-22-02823-f013:**
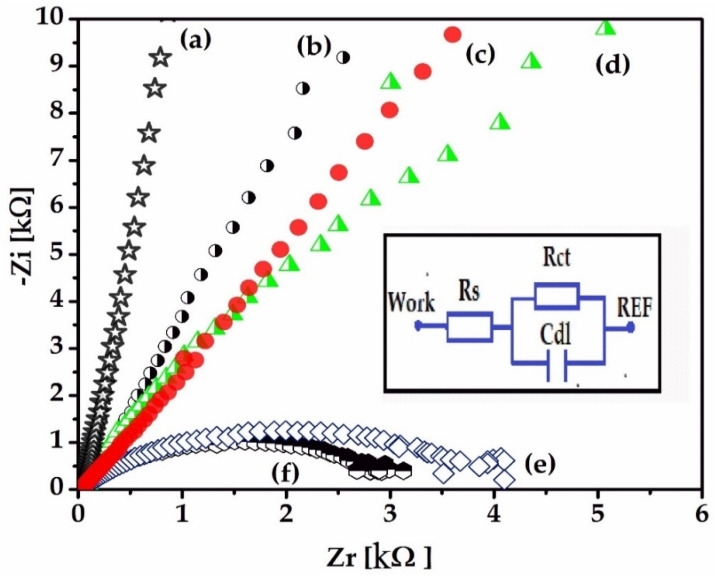
Nyquist diagrams (-Zi vs. Zr) taken over time for the electrode permanently immersed in the PBS 7.1 electrolyte solution for a frequency range between 100 mHz and 100 kHz at: 0 days (a); 1st day (b); 2nd day (c); 7th day (d); 14th day (e); 21st day (f). Inset: the equivalent circuit with: resistance charge transfer (Rct), capacitive dielectric double layer (Cdl), and resistance solution (Rs).

**Figure 14 sensors-22-02823-f014:**
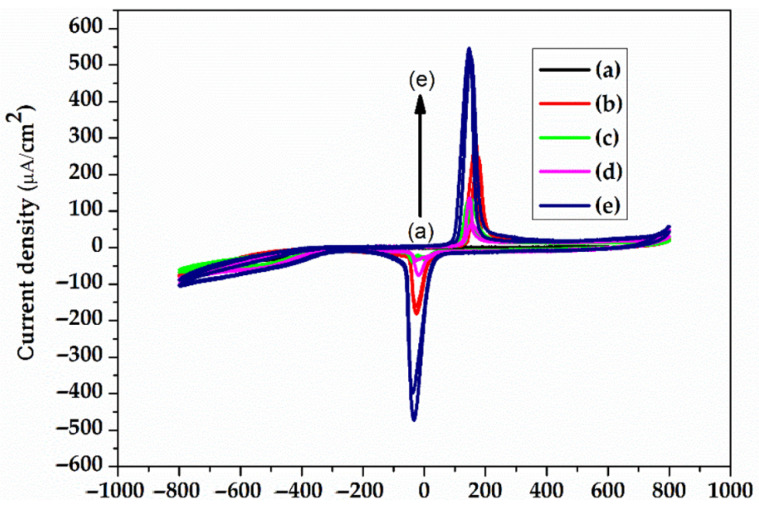
CVs (−800 mV–800 mV) of the electrodes in the electrolyte solution PBS 7.1 after: 0 day (a); 1st day (b); 7th day (c); 14th day (d); and 21st day (e).

**Figure 15 sensors-22-02823-f015:**
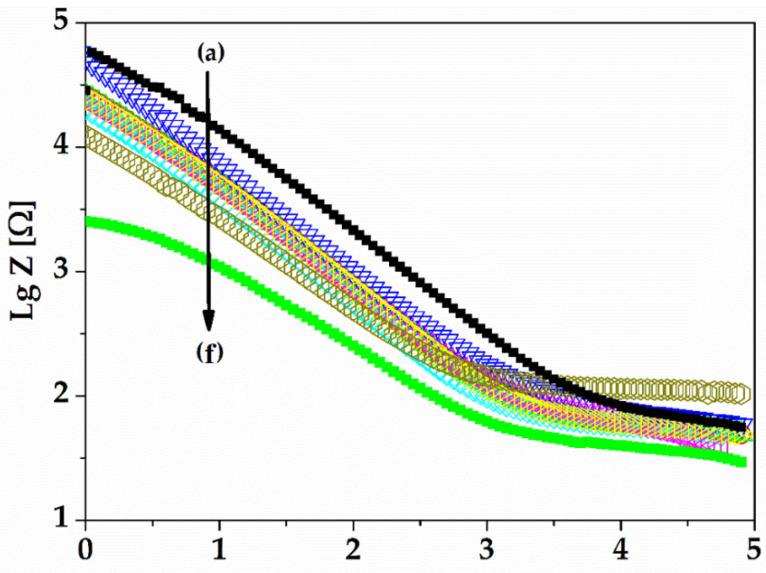
Electrochemical impedance spectra of gold electrodes over 21 days of experiments, as a function of frequency: 0 day (a); 1st day (b); 2nd day (c); 7th day (d); 14th day (e); and 21st day (f).

**Figure 16 sensors-22-02823-f016:**
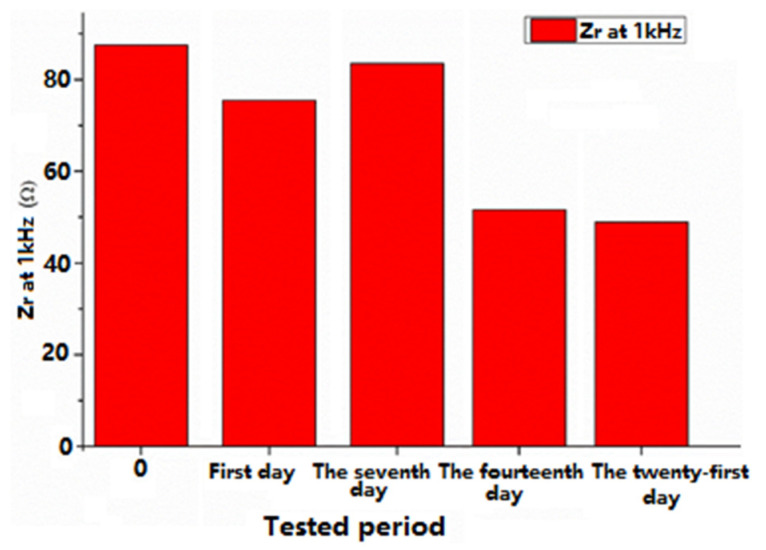
Real impedance vs. time at 1 kHz.

**Figure 17 sensors-22-02823-f017:**
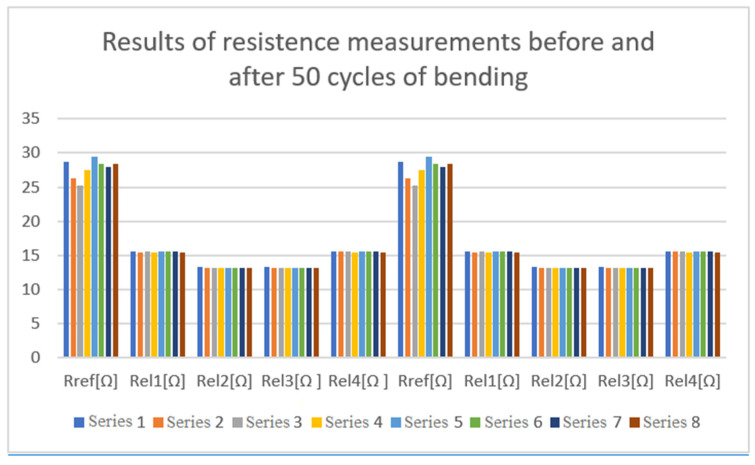
Electrode resistance before and after bending.

**Figure 18 sensors-22-02823-f018:**
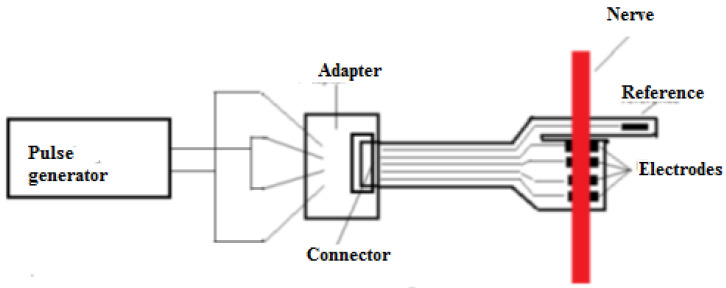
Block diagram of the testing circuit.

**Figure 19 sensors-22-02823-f019:**
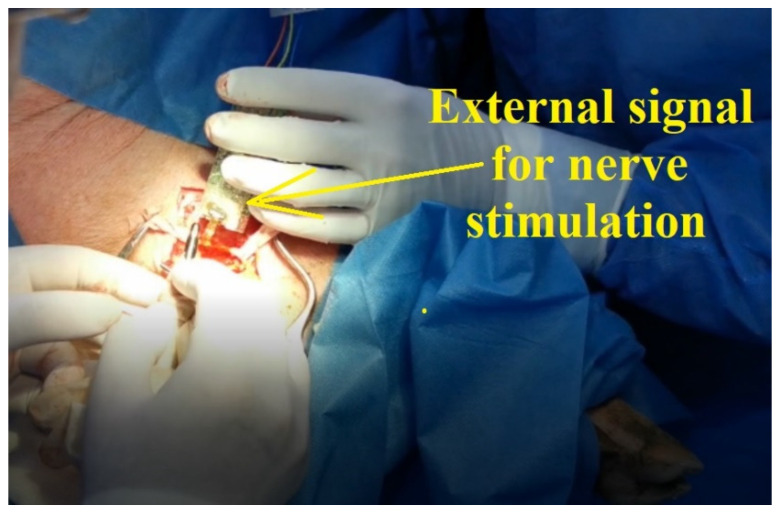
Pulse generator used to induce movement of swine leg.

**Figure 20 sensors-22-02823-f020:**
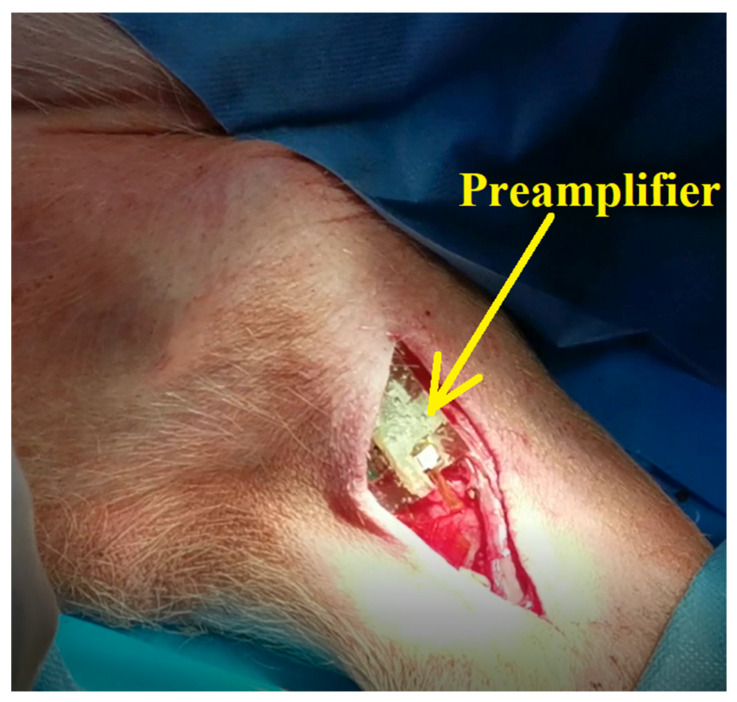
Installing the preamplifier (front-end electronics).

**Figure 21 sensors-22-02823-f021:**
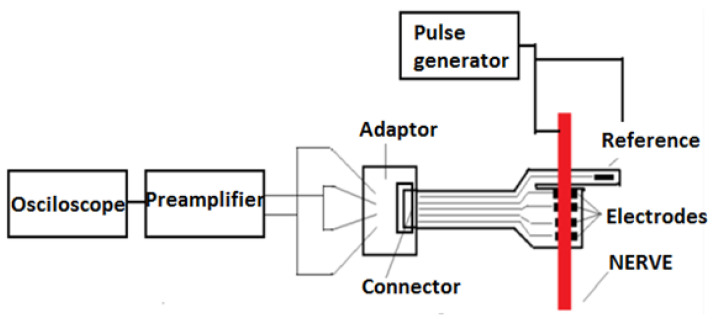
The block diagram of the experiment.

**Figure 22 sensors-22-02823-f022:**
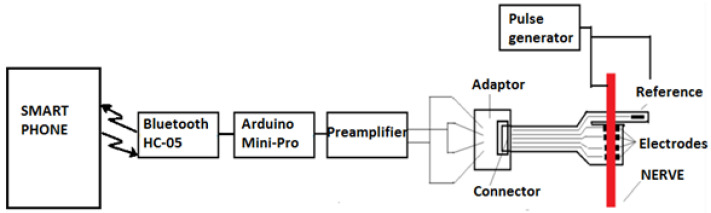
The block diagram of the electronic test module.

**Figure 23 sensors-22-02823-f023:**
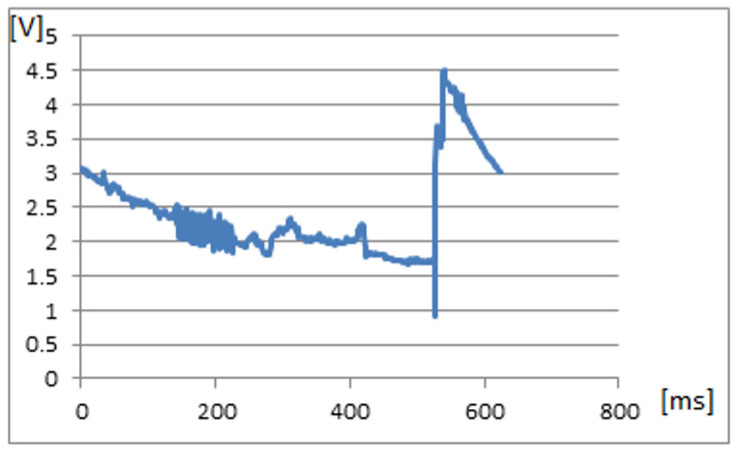
The signal recorded during the nerve stimulation.

**Figure 24 sensors-22-02823-f024:**
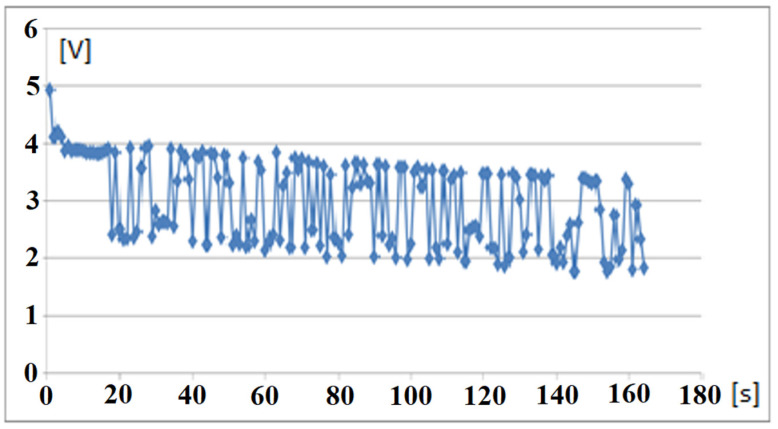
The string of signals received on the mobile phone from the Bluetooth transmitter upon stimuli applied to nerve.

**Table 1 sensors-22-02823-t001:** Measurements of electrode resistance before and after bending.

	Before Bending					After 3 Cycles of 50 Bendings Each				
Sample	R_ref_ [Ω]	R_el1_ [Ω]	R_el2_ [Ω]	R_el3_ [Ω ]	R_el4_ [Ω ]	R_ref_ [Ω]	R_el1_ [Ω]	R_el2_ [Ω]	R_el3_ [Ω]	R_el4_ [Ω]
1	28.73	15.60	13.21	13.21	15.60	28.73	15.60	13.21	13.21	15.60
2	26.24	15.46	13.19	13.19	15.56	26.24	15.46	13.19	13.19	15.56
3	25.22	15.49	13.20	13.20	15.49	25.22	15.49	13.20	13.20	15.49
4	27.54	15.46	13.17	13.17	15.46	27.54	15.46	13.17	13.17	15.46
5	29.51	15.60	13.19	13.19	15.60	29.51	15.60	13.19	13.19	15.60
6	28.38	15.49	13.20	13.20	15.49	28.38	15.49	13.20	13.20	15.49
7	27.88	15.48	13.18	13.18	15.48	27.88	15.48	13.18	13.18	15.48
8	28.34	15.45	13.17	13.17	15.45	28.34	15.45	13.17	13.17	15.45

## Data Availability

Not applicable.

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
