# Peer review of "Remote Sensing System for Motor Nerve Impulse"

_sensors, 2022, doi:10.3390/s22082823_

Round 1

Reviewer 1 Report

Authors have presented a system developed for sensing motor nerve impulse and demonstrated the system in a porcine model. Overall the paper is presented with adequate details on the system development and testing. Results mostly have also been corroborated by the data presented. A claim made in the paper about the “main original contribution” of the paper regarding “the electrode pair and the reference electrode” is questionable, a similar electrode system was demonstrated in a publication by the same authors (K. Imenes et al., 2021). Is there a specific reason for the article not to be mentioned? Why was a different electrode material consider from previous study? Additionally, the authors point out that there is limitation in deploying the technology due to corrosion. The claim that polymeric components that could help the longevity of the electrodes is not tested.

The following are additional questions that need to be addressed:

  1. How is the BLE system better/different from the 433MHz transmitter used in the previous literature (K. Imenes et al., 2021)?
  2. Figure 2 The text describes a BLE system but the figure is labelled as WIFI. Can you clarify which communication technique is used? The overall quality of the image is low try to replace with better images.
  3. Quality of the images are low throughout the paper. Please try to present the data using a better plotting software. Figure 10 is not necessary, instead include a picture of the resistance measurement on the electrode.
  4. Why is a 2-stage amplification required? INA118 can have a gain of 10000 why not have a single amplification stage? Why is the current feedback circuitry bandwidth important?
  5. The in vitro test described is not a biocompatibility test, it is a cytocompatibility/cytotoxicity test. By the definition of biocompatibility establishing it would require more tests such as sensitization, cytotoxicity and hemacompatibilty to name a few. The testing method is also not clear, try to rephrase the text for clarity. Number of samples in the study seems too low, was any analysis done to validate the sample size? Why is the power set at 0.1?
  6. The electrodes seem to be under repeated strain rather than high stress described in section 2.5. Also, illustrate this testing method to show how and what points the stress was applied.
  7. Is SD standard deviation?
  8. Table 1 can be combined into a graph.
  9. Not clear what figure 22 and 23 means, what is series 1? Sampling rate in figure 23 seems to be low, is this rate sufficient/comparable to other sensing systems?

K. Imenes et al., "Implantable Interface for an Arm Neuroprosthesis," 2021 23rd European Microelectronics and Packaging Conference & Exhibition (EMPC), 2021, pp. 1-8, doi: 10.23919/EMPC53418.2021.9585011.

Author Response

Thank you for the reviewers’ comments concerning our paper entitled “Remote Sensing System for Motor Nerve Impulse”. Your comments are very helpful for revising and improving our paper, as well as the important guiding significance to our research. Now, we are submitting the revised version after seriously considering the comments of the Reviewers. We hope that this version will improve the quality of our manuscript and make it more acceptable for publication.

On behalf of the co-authors, I would like to clarify some of the points brought up by the Reviewers. We hope that the Reviewers and the Editor will be satisfied with our replies to the comments and our manuscript.

Sincerely,

Carmen Moldovan

Reviewer I

Authors have presented a system developed for sensing motor nerve impulse and demonstrated the system in a porcine model. Overall, the paper is presented with adequate details on the system development and testing. Results mostly have also been corroborated by the data presented. A claim made in the paper about the “main original contribution” of the paper regarding “the electrode pair and the reference electrode” is questionable, a similar electrode system was demonstrated in a publication by the same authors (K. Imenes et al., 2021). Is there a specific reason for the article not to be mentioned? Why was a different electrode material consider from previous study? Additionally, the authors point out that there is limitation in deploying the technology due to corrosion. The claim that polymeric components that could help the longevity of the electrodes is not tested.

Thank you very much for this indication. It was our mistake. We added this reference at the very beginning (reference no 2) and removed these claims from the conclusions. As was mentioned in the previous article [K. Imenes et al., 2021], we fabricated two types of electrodes (using PDMS and Kapton). The technological challenges (with PDMS and Kapton) were very different, and the previous article is entirely dealing with the PDMS version. We are also exploring the Kapton technology because our medical team consider the implanting procedure depends on the substrate material. At this point it seems that the Kapton substrate might be preferred by the doctors because can it by easily wrapped -up around the nerve, because is very thin (25microns) and extremely flexible.  It is also preferred by the technologists because of the excellent adhesion of the metal layer Ti/Au used as conductive later of the electrodes.  We have adjusted the Conclusion sections.

The following are additional questions that need to be addressed:

  1. How is the BLE system better/different from the 433MHz transmitter used in the previous literature (K. Imenes et al., 2021)?

We took the decision of switching to the Bluetooth system from a practical reason: We had to collect data from our module implanted on the swine and it was much easier to enter the stable with a telephone and connect to the electronic module; however, both solutions were developed within the ARMIN project.

  1. Figure 2 The text describes a BLE system but the figure is labelled as WIFI. Can you clarify which communication technique is used? The overall quality of the image is low try to replace with better images.

We used BLE and the figure was adjusted.

  1. Quality of the images are low throughout the paper. Please try to present the data using a better plotting software. Figure 10 is not necessary, instead include a picture of the resistance measurement on the electrode.

Several figures were improved: figure 2, figure 8, figure 10, figure 11, figure 19, figure 20, figure 23, figure 24.

We added a figure with the electrodes during the measurement of the mechanical resistance

  1. Why is a 2-stage amplification required? INA118 can have a gain of 10000 why not have a single amplification stage? Why is the current feedback circuitry bandwidth important?

We estimated a stimulus with an amplitude of 5mV, however, after discussing with doctors and become aware that in time it is possible that conductivity of myeline cover of the axon could decrease over time and consequently we decided to use a chain of two amplifiers and to establish a control loop on the second amplifier in order to compensate this variation.

  1. The in vitro test described is not a biocompatibility test, it is a cytocompatibility/cytotoxicity test. By the definition of biocompatibility establishing, it would require more tests such as sensitization, cytotoxicity and hemacompatibilty to name a few. The testing method is also not clear, try to rephrase the text for clarity. Number of samples in the study seems too low, was any analysis done to validate the sample size? Why is the power set at 0.1?

We have updated the section in the article according to your remarks (cytocompatibility/cytotoxicity test).  The text is rephrased, and is now clearer on the testing method.

The number of low samples is due to the limited number of available samples at the time of testing. No analysis is done to validate the sample size itself.

Common power or significance levels used are 0.01, 0.05 and 0.1; with 0.05 as the most used one. In our case we use a significance level of 0.1. This is due to the localized effect of the copper wire (used as negative control) which is a result of the physical size of the negative control sample. Using a physically larger negative control sample, would probably give a more overall effect, and a lower significance level could possibly have been applied e.g. 0.05.

  1. The electrodes seem to be under repeated strain rather than high stress described in section 2.5. Also, illustrate this testing method to show how and what points the stress was applied.

Stress and strain are closely related, as strain can result from applied stress. Nevertheless, we have changed the text to clarify.

A new figure had be added in section 2.5 showing at which points the electrode are attached to the instrument MECMESIN Multittest. The stress would be applied at these points, and resulting in a strain in the electrode.

  1. Is SD standard deviation?

Yes; SD is standard deviation. And we had several typos, 106 instead of 106

Mean cell counts were 1.61 x 106 cells/ml (SD 4.8 x 105 cells/ml) for positive control; 6.2 x 105 cells/ml (SD 1.06 x 105 cells/ml) for negative control; and 1.24 x 106 cells/ml (SD 1.71 x 105 cells/ml)

  1. Table 1 can be combined into a graph.

We introduced a graph as you suggested (Figure 17)

  1. Not clear what figure 22 and 23 means, what is series 1? Sampling rate in figure 23 seems to be low, is this rate sufficient/comparable to other sensing systems?

In figure 22 we have the recorded result for one stimulus while in figure 23 is the result of a sequence of stimuli applied on the nerve. 

Reviewer 2 Report

This manuscript provides experimental evidence for a new sensing system developed for nervous pulse acquisition. Some electrical, chemical, and mechanical tests are conducted on the implantable device. However, there are still parts of the manuscript requiring further characterization. Here are some:

  • The results suggest electrode deterioration happening over a long-term usage of the device. How does this then translate to its long-term efficacy, especially for the case of a wide range of working frequencies and voltages?
  • Experimental results similar to what shown in Figs. 22 and 23, but for the case of running the device for over a few weeks can be included to be able to verify the effectiveness of this device for long-term applications.
  • What is the sensitivity of the device in detecting low amplitude stimuli?
  • Why 1 kHz stimulus has been chosen for running the tests?

Minor comments:

  • Please check the text for any typos/grammatical errors. E.g., “In addition, by using these modules is providing …”; “The electrolyte solution used was PBS (phosphate buffered saline), 7.1 in deionized water, suitable for the pH of the physiological environment in which the electrode is implanted.”
  • Some figures, including Fig. 8a and 10, can be moved to supporting information text, as they are not scientifically contributing to the main message of the manuscript.
  • Figure 11 is not informative. The two arrows seem to be pointing to similar locations where it is hard to distinguish nerve from electrodes.
  • Figure 18 is not informative either. How can the reader observe the movement of the leg in a snapshot in time?! A video can communicate the message much better.
  • In Fig. 19, it is hard to see where exactly the “Implanted electrodes” are.
  • Figures 17, 20, and 21 can be moved to supporting information.
  • What does “Series 1” in Figures 22 and 23 refer to? The two figures can be combined. Figure 23 has a very low resolution, which needs to be replaced with a higher-quality one.

Author Response

Dear Editor and Reviewers,

Thank you for the reviewers’ comments concerning our paper entitled “Remote Sensing System for Motor Nerve Impulse”. Your comments are very helpful for revising and improving our paper, as well as the important guiding significance to our research. Now, we are submitting the revised version after seriously considering the comments of the Reviewers. We hope that this version will improve the quality of our manuscript and make it more acceptable for publication.

On behalf of the co-authors, I would like to clarify some of the points brought up by the Reviewers. We hope that the Reviewers and the Editor will be satisfied with our replies to the comments and our manuscript.

Sincerely,

Carmen Moldovan

Reviewer II

This manuscript provides experimental evidence for a new sensing system developed for nervous pulse acquisition. Some electrical, chemical, and mechanical tests are conducted on the implantable device. However, there are still parts of the manuscript requiring further characterization. Here are some:

Thank you for your help to improve our paper. We have included in the article new information in order to address all your suggestions, as mentioned below:

  • The results suggest electrode deterioration happening over a long-term usage of the device. How does this then translate to its long-term efficacy, especially for the case of a wide range of working frequencies and voltages?

We are in the stage of development and our teams is working to further improve this aspect. However, in the real application (forearm prothesis), with the human patient, the electrodes are not exposed at the same stress as they were in the animal experiment.

  • Experimental results similar to what shown in Figs. 22 and 23, but for the case of running the device for over a few weeks can be included to be able to verify the effectiveness of this device for long-term applications.

We are aware of the necessity of long-term experiments, but we did not have the opportunity and resources so far to proceed in this direction. According to our research project timeline, we will do this type of experiments in the future. We have included this consideration in the Conclusions, as a future direction of research.

  • What is the sensitivity of the device in detecting low amplitude stimuli?

The INA 118 has the maximum offset as low as 50 µV according to the datasheet [1].

  • Why 1 kHz stimulus has been chosen for running the tests?

We have studied the literature and we discussed also with the doctors this issue. ”Physiologically, action potential frequencies of up to 200-300 per second (Hz) are routinely observed. Higher frequencies are also observed, but the maximum frequency is ultimately limited by the absolute refractory period. Because the absolute refractory period is ~1 ms, there is a limit to the highest frequency at which neurons can respond to strong stimuli. That is to say that the absolute refractory period limits the maximum number of action potentials generated per unit time by the axon. As described previously, the strength of the stimulus must be very high in order to ensure that the duration of the action potential is as short as the duration of the absolute refractory period. A stronger than normal stimulus is required to overcome the relative refractory period. Because the absolute refractory period can last between 1-2 ms, the maximum frequency response is 500-1000 s−1 (Hz). In order to be confident that the developed system is working well we decided to use the upper limit -1kHz[2]”. This reference was included in the paper.

 [1] INA118 Precision, Low-Power Instrumentation Amplifier, Texas instruments, SBOS027B –September 2000–Revised April 2019, Available online: https://www.ti.com/lit/ds/symlink/ina118.pdf

[2] Neuronal Action Potential - Frequency Coding in the Nervous System, Physiologyweb, 2022, Available online: https://www.physiologyweb.com/lecture_notes/neuronal_action_potential/neuronal_action_potential_frequency_coding_in_the_nervous_system.html

 Minor comments:

  • Please check the text for any typos/grammatical errors. E.g., “In addition, by using these modules is providing …” replaced with: ”In addition, by using these modules is provided ..”

“The electrolyte solution used was PBS (phosphate buffered saline), 7.1 in deionized water, suitable for the pH of the physiological environment in which the electrode is implanted.” was replaced with: ”The electrolyte solution used was pH 7.1 PBS (phosphate buffered saline), the standard solution for physiological medium experiments”.

We have done the corrections, thank you.

  • Some figures, including Fig. 8a and 10, can be moved to supporting information text, as they are not scientifically contributing to the main message of the manuscript.

Figure 8b was added in order to explain the bending cycles.

Figure 10a with the measurement of the electrodes resistance was added.

  • Figure 11 is not informative. The two arrows seem to be pointing to similar locations where it is hard to distinguish nerve from electrodes.

Figure 11 was improved.

  • Figure 18 is not informative either. How can the reader observe the movement of the leg in a snapshot in time?! A video can communicate the message much better.

In figure 18 was removed the comment with the movement of the leg.

  • In Fig. 19, it is hard to see where exactly the “Implanted electrodes” are.

The main topic of figure 19 is the implanted electronic module. The electrodes indeed are not visible, this text was deleted.

  • Figures 17, 20, and 21 can be moved to supporting information.

  • What does “Series 1” in Figures 22 and 23 refer to? The two figures can be combined. Figure 23 has a very low resolution, which needs to be replaced with a higher-quality one.

In figure 22 we have the recorded result for one stimulus while in figure 23 is the result of a sequence of stimuli applied on the nerve. Figure 23 was replaced with a better-quality one (the figures were renumbered).

Several images were replaced with better quality similar ones: figure 2, figure 8, figure 10, figure 11, figure 19, figure 20, figure 23, figure 24.

Round 2

Reviewer 2 Report

The authors have implemented/addressed the comments from the first round of review to some extent. However, some points could be addressed more directly. For instance, the lowest sensitivity of the device in operation in vivo has been claimed to be 50 uV but no data has been shown to support this. 

Author Response

Thank you for the reviewers’ comments concerning our paper entitled “Remote Sensing System for "Motor Nerve Impulse”. Your comments are very helpful for revising and improving our paper, as well as the important guiding significance to our research. Now, we are submitting the revised version after
seriously considering the comments of the Reviewers. We hope that this version will improve the quality of our manuscript and make it more acceptable for publication.

Comment/sugestion

The authors have implemented/addressed the comments from the first round of review to some extent. However, some points could be addressed more directly. For instance, the lowest sensitivity of the device in operation in vivo has been claimed to be 50 uV but no data has been shown to support this. 

Answer 

The INA 118 features a maximum offset voltage of 50 µV and a gain of 1000[1]. While our in vivo objective was a 5 mVpp amplitude stimulus, due to the foreign body reaction creating scar tissue to raise impedance over time [2],[3],[4]  and the amplitude of the input signal could decrease accordingly. In order to develop a versatile front end for the used electrodes we decided to choose a gain of 100 for the INA 118 and to use a chain of two amplifiers in order to further establish a control loop on the second amplifier, in order to be able to compensate this variation. We are looking forward to perform a long-term test as soon as the solution for wireless charging battery will be developed and then, based on the obtained data we could increase the gain of INA 118 if necessary or compensate the impedance modification only by maintaining the already established control loop. We were looking also for an initial solution providing flexibility and which is easy to be modified without changing the PCB design or in the worst case a complete modification of electronic design.  

[2] Salatino, J.W., Ludwig, K.A., Kozai, T.D.Y. et al. Glial responses to implanted electrodes in the brain. Nat Biomed Eng 1, 862–877 (2017). https://doi.org/10.1038/s41551-017-0154-1

[3]  Lotti F, Ranieri F, Vadalà G, Zollo L and Di Pino G (2017) Invasive Intraneural Interfaces: Foreign Body Reaction Issues. Front. Neurosci. 11:497. doi: 10.3389/fnins.2017.00497

[4] Sridharan A, Rajan SD, Muthuswamy J. Long-term changes in the material properties of brain tissue at the implant-tissue interface. J Neural Eng. 2013 Dec;10(6):066001. 
